# Non-Destructive Evaluation of Steel Surfaces after Severe Plastic Deformation via the Barkhausen Noise Technique

**Miroslav Neslušan [1,\*], Libor Trško [1] , Peter Minárik [2] , Jiří Čapek [3], Jozef Bronček [1], Filip Pastorek [1], Jakub Čížek [4] and Ján Moravec [1]**

[1]  University of Žilina, Univerzitná 8215/1, 010 26 Žilina, Slovakia; libor.trsko@rc.uniza.sk (L.T.); jozef.broncek@fstroj.uniza.sk (J.B.); filip.pastorek@rc.uniza.sk (F.P.); jan.moravec@fstroj.uniza.sk (J.M.)

[2]  Faculty of Mathematics and Physics, Charles University, Ke Karlovu 5, 121 16 Praha 2, Czech Republic; peter.minarik@mff.cuni.cz

[3]  Faculty of Nuclear Sciences and Physical Engineering, ČVUT Praha, Trojanova 13, 120 00 Praha 2, Czech Republic; jiri.Capek@fjfi.cvut.cz

[4]  Faculty of Mathematics and Physics, Charles University, V Holešovičkach 2, 180 00 Praha 8, Czech Republic; jcizek@mbox.troja.mff.cuni.cz

[\*]  Correspondence: miroslav.neslusan@fstroj.uniza.sk, Tel.: +421-908-811-973

**Abstract:** This paper reports about the non-destructive evaluation of surfaces after severe shot peening via the Barkhausen noise technique. Residuals stresses and the corresponding Almen intensity, as well as microstructure alterations, are correlated with the Barkhausen noise signal and its extracted features. It was found that residual stresses as well as the Barkhausen noise exhibit a valuable anisotropy. For this reason, the relationship between the Barkhausen noise and stress state is more complicated. On the other hand, the near-the-surface layer exhibits a remarkable deformation induced softening, expressed in terms of the microhardness and the corresponding crystalline size. Such an effect explains the progressive increase of the Barkhausen noise emission along with the shot-peening time. Therefore, the Barkhausen noise can be considered as a promising technique capable of distinguishing between the variable regimes of severe shoot peening.

**Keywords:** shot peening; Barkhausen noise; crystallite size

## 1. Introduction

Conventional shot peening (CSP) is a widely employed technique applied for the final surface processing of components, which improves the mechanical properties and the corresponding fatigue or/and corrosion behavior under applied stress [1–4]. Shot peening (SP) usually alters the surface morphology; near surface microstructure expressed in terms of microhardness, grain size, and/or dislocation density; and the stress state [1–4]. The CSP surfaces contain compressive residual stresses of a variable magnitude and penetration depth. These stresses strongly correspond with the Almen intensity as a parameter that can be easily measured on the commonly used Almen strips. The Almen intensity refers to the arc height of the Almen strip after shot peening, and the required Almen intensity can be obtained by adjusting the SP conditions, such air pressure, shot size, peening time, and others. Unal [5] reports that the Almen intensity is directly considered as a major causal agent of all of the mechanical and metallurgical changes in the surface.

Severe shot peening (SSP) is usually employed when a higher magnitude of compressive stresses and higher degree of microstructure alterations are required (especially grain refinement). For these reasons, higher Almen intensities and degrees of coverage are required for the SSP than those for

the CSP. Based on the value of the exerted kinetic energy (except CSP and SSP), the over and re-shot peening processes can be employed as well. The shot peening process is widely reported from many points of view, and published studies discuss a variety of shot peened materials under variable conditions. Kleber and Barroso [1] reported about strain-induced martensite transformations of austenitic steel AISI 304L of a different extent and degree under the free surface, using the variable surface coverage during the CSP. Unal and Varol [2] also refer to the mechanical twins in AISI 304 after the CSP, SSP, and re-peening. Fargas et al. [6] and Chen et al. [7] clearly showed that extensive plastic deformation initiates phase transformations in duplex steel when the fraction of martensite increases at the expense of austenite. Moreover, the CSP process also remarkably increases the dislocation density and decreases the domain size in the near surface region. Fu et al. [3] analyzed the surface hardness and stress state after the CSP and the consecutive annealing. Unal [5] carried out an optimization of the Almen intensity, surface roughness, and hardness, varying the input processing parameters. Segurado et al. [8] investigated the effect of different types of shots on surface morphology, residual stresses, and fatigue behaviour. Maleki et al. [9] studied the influence of various CSP regimes on grain refinement, stress, and microhardness profiles, as well as on S-N curves. Maleki and Unal [10] studied the influence of surface coverage and the re-peening process on the properties of AISI 1045 steel after CSP as well as SSP. Trško et al. studied the CSP and SSP process in relation to the fatigue properties of steel [11] and aluminium alloys [12], together with influence on the surface texture and fracture surface character [13]. His results showed that the SSP process can be more beneficial to the fatigue life of a material, however, exceeding a certain point leads to significant surface damage and a rapid drop of fatigue properties.

The CSP and SSP alter the surface integrity in the complexity of this term. The surface state after the SP is affected by many input variables. Therefore, the SP cycle performed on the real components requires the preliminary phase when the SP parameters are adjusted using the Almen strips. On the other hand, the surface state of the real components after SP can vary, especially when the components of a complicated geometry are shot peened, even when the Almen strips are mounted directly on the component in various places. For this reason, the non-destructive technique would be beneficial for monitoring such components in the real production. Magnetic Barkhausen noise (MBN) is a magnetic technique based on an irreversible and discontinuous domain wall motion. The domain wall motion and the corresponding MBN signal (as well as the extracted MBN parameters) are sensitive to the stress state and to the microstructure alterations. The stresses affect mainly the alignment of the domain walls, whereas the microstructure features affect the pinning strength of the matrix and the free path of the motion of the domain walls. It is well known that a higher MBN can be obtained under tensile stresses rather than compressive ones [14–19]. Furthermore, the domain wall motion can be strongly pinned by dislocation cells [20,21], precipitates [22], and/or non-ferromagnetic phases [23]. The concept in which the SP surface could be accessed via the MBN technique has been already published. Kleber and Barroso [1] found that the degree of strain-induced martensite transformation in austenitic steel AISI 304L strongly correlates with the MBN and extracted MBN envelopes. Theiner and Hauk [24] reported that macro and micro stresses, the work hardening depth, and the homogeneity of the CSP surface can be determined using the MBN parameters. Sorsa et al. [25], Gur and Savas [26], and Tiitto and Francino [27] correlated the CSP parameters to the residual stress profiles and MBN. Marconi et al. [28] employed the CSP for the hydraulic Pelton wheels in order to increase their fatigue resistance. The authors found that the CSP drastically influences the magnetic properties of the surface and correlates MBN with the stress state and volume of the retained austenite.

As it has been reported, the potential of MBN technique for the SP surface has been investigated for many years. Compared to CSP, SSP represents the production of excessively peened surfaces containing a more developed matrix alteration, especially in the near surface region and in the deeper extent of the compressive stresses. There is no exact boundary between the CSP and SSP treatments, because it significantly depends on the type of the treated material. For example, the SSP parameters for steels are much more severe than for aluminum alloys. However, the CSP process is mainly

connected with accumulation of residual stresses in the subsurface volume, while the SSP treatment is considered when, besides the compressive residual stresses, significant changes in the microstructure (mainly grain refinement) occur. For these reasons, the magnetic properties and the corresponding MBN emission (due to the superimposing effects of the microstructure and residual stresses alterations) after SSP should also be affected. This is a case study focused on the SSP surfaces obtained via different regimes (the corresponding Almen intensities), in which the influence of the residual stresses and microstructure on the MBN emission is investigated.

## 2. Materials and Methods

Experiments were carried out on Almen A strips made of AISI 1070 spring steel (cold rolled, tensile strength 640 MPa, yield strength 495 MPa, strips of thickness $1.295 \pm 0.025$ mm, width $19.05 \pm 0.127$ mm, and length $76.2 \pm 0.381$ mm, shot peening center of Polytechnic University of Milan, Italy). Heat treatment: oil quenched from a heating temperature of 810 °C $\pm$ 20 °C, and subsequently high tempered at 440 °C for hardness 44 ÷ 45 HRC. The chemical composition is indicated in Table 1. Strips length is referred in the rolling direction (RD), whereas the width of the strips is referred to as the transversal direction (TD). The SSP parameters are specified in Table 2. Figure 1 illustrates the evolution of the Almen intensity for the different shot peening regimes. The measurements were carried out on the deformed Almen strips. The shots (S170) hardness occurs in the range of 40 to 51 HRC, whereas the strips hardness is $468 \pm 22$ Vickers microhardness (HV), which corresponds $46 \pm 2$ HRC.

The surface roughness was measured using the Hommel Tester T 2000 in RD (three repetitive measurements on each strip, length 12.5 mm in length, Hommel Werke, Jena, Germany). To reveal the microstructural or other transformations induced by the SSP, 10 mm long pieces were routinely prepared for the SEM observations (hot molded, ground, polished, and etched by 1% Nital for 10 s). The microstructure was observed in RD.

Vickers microhardness (HV) testing was conducted using a Zwick Roel ZHm microhardness tester (Zwick Roell, Ulm, Germany) by applying a force of 50 g for 10 s. The microhardness was determined by averaging three repeated measurements (three microhardness profiles spaced at 0.15 mm). The bulk microhardness was measured at a depth of approximately 0.5 mm below the free surface.

The MBN was measured by using a RollScan 350 (Stresstech, Jyväskylä, Finland), and was analyzed using MicroScan 500 software (magnetizing voltage 7.5 V, magnetizing frequency 125 Hz, sensor type S1-18-12-01, frequency range of MBN pulses in the range from 10 to 1000 kHz). The MBN values were obtained by averaging ten MBN bursts (five magnetizing cycles). The MBN refers to the root mean square (RMS) value of the signal. Taking into consideration the hardness of the samples, the estimated sensing depth of the MBN signal is ~70 µm [29]. The magnetization of the strips was carried out in the RD and TD directions. In addition to the conventional MBN parameter (RMS value of the signal), the peak position (PP) of the MBN envelope was also analyzed.

**Table 1.** Chemical composition of the AISI 1070 steel in wt%.

| Fe | C | Mn | P | S |
|---------|------|------|-----------|-----------|
| balance | 0.70 | 0.75 | max. 0.04 | max. 0.05 |

**Table 2.** Shot peening parameters.

| Shot Peening Time (s) | Shot Size (mm-S170) | Almen Intensity (mm A) | *Ra* (µm) | *Rz* (µm) |
|---|---|---|---|---|
| 10 | 0.4318 | 14.1 | $4.41 \pm 0.20$ | $31.53 \pm 2.50$ |
| 20 | 0.4318 | 15.6 | $4.70 \pm 0.18$ | $34.00 \pm 1.73$ |
| 40 | 0.4318 | 15.8 | $4.59 \pm 0.14$ | $33.02 \pm 4.01$ |
| 80 | 0.4318 | 16.6 | $5.48 \pm 0.41$ | $37.67 \pm 3.06$ |

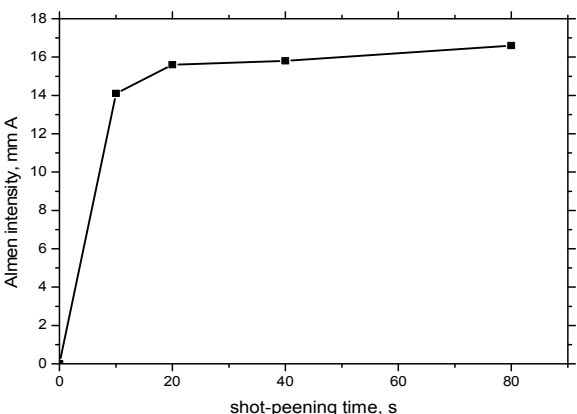

**Figure 1.** Evolution of Almen intensities with the shot peening time.

The determination of the residual stresses was performed using the XRD technique carried out on a Proto iXRD Combo diffractometer (Proto manufacturing Ltd., Ontario, Canada) using Cr$K_\alpha$ radiation. The average effective penetration depth of the XRD measurements was ~5 μm, the scanning angle was ±39°, and the Bragg angle was 156.4°. The residual stress was calculated from the shifts of the 211 reflection. The Winholtz and Cohen method and X-ray elastic constants $\frac{1}{2}s_2$ = 5.75 TPa$^{-1}$ and $s_1$ = −1.25 TPa$^{-1}$ were used for the residual stress determination. In order to analyze the stress gradients beneath the surface of the samples, layers of material were gradually removed using electro-chemical polishing in the center of the sample. The crystallite size (the size of coherently diffracted domains) and dislocation density $\rho_D$ were determined using the XRD patterns obtained on a X'Pert PRO diffractometer (Panalytical Ltd., Eindhoven, Netherlands), using the Co$K_\alpha$ radiation and subsequent Rietveld refinement performed on Mstruct software (version 0.1, Charles University, Czech Republic) [30–34] (used for the fitting of the measured XRD patterns). The effective penetration depth of the XRD measurements was, in this case, 1 ÷ 5 μm. Using the XRD data, the crystallite size (*t*) was calculated via the Scherrer formula (Equation (1)), as follows:

$$t = \frac{K \cdot \lambda}{B \cdot \cos\theta} \tag{1}$$

where *K*, *λ*, *B*, and *θ* are the shape factor, X-ray wavelength, peak broadening and half of the diffraction angle, and the Bragg angle, respectively.

A brief sketch of the analyses' positioning on the strips (as well as their order) is illustrated in Figure 2. The first phase represents non-destructive measurements, followed by the XRD stress profile investigations. Consequently, the shot peened strips were cut and hot molded for SEM observations and microhardness measurements.

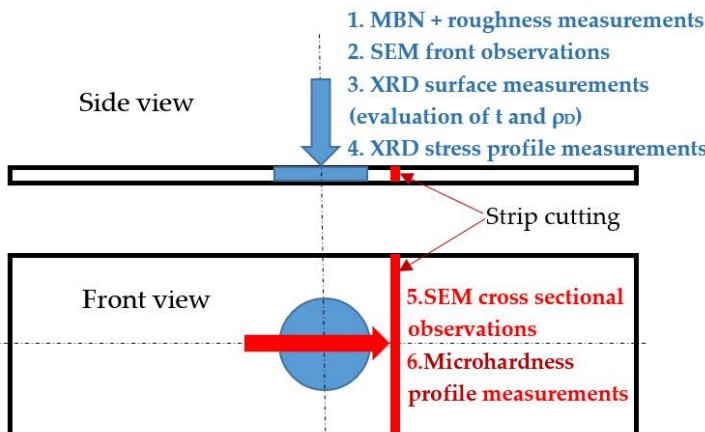

**Figure 2.** Brief sketch of experimental setup. MBN—Magnetic Barkhausen noise.

## 3. Results

### 3.1. SEM Observations

The cross sectional images after SSP (Figure 3) exhibited no distinctive difference in the microstructure for all of the shot peening times. The deeper regions remain untouched by the SSP process, and the near surface layer does not contain indications of remarkable structure transformations and/or severe plastic deformation.

On the other hand, the front view of the SEM images of the shot peened surfaces (Figure 4) illustrate that the original surface was remarkably notched by cold rolling (see Figure 4a). This is particularly visible in the areas not covered by the shot peening process after the 10 s treatment, and the notches are preferentially oriented in the rolling direction. A further increase in the shot peening time, and consequently, an increase in the surface coverage produces a surface fully covered by shot peening dimples. However, the notches that originated from cold rolling are still preserved on the surface (except SSP time 80 s), but tend to diminish with the increasing shot peening time. Such behavior is associated with the increasing intensity of plastic deformation as a result of the repetitive effect of shot impacts and the accumulation of their energy in the surface. Finally, the surface shot peened for 80 s contains no notches; however, it seems to be over peened and contained multiple scurfs.

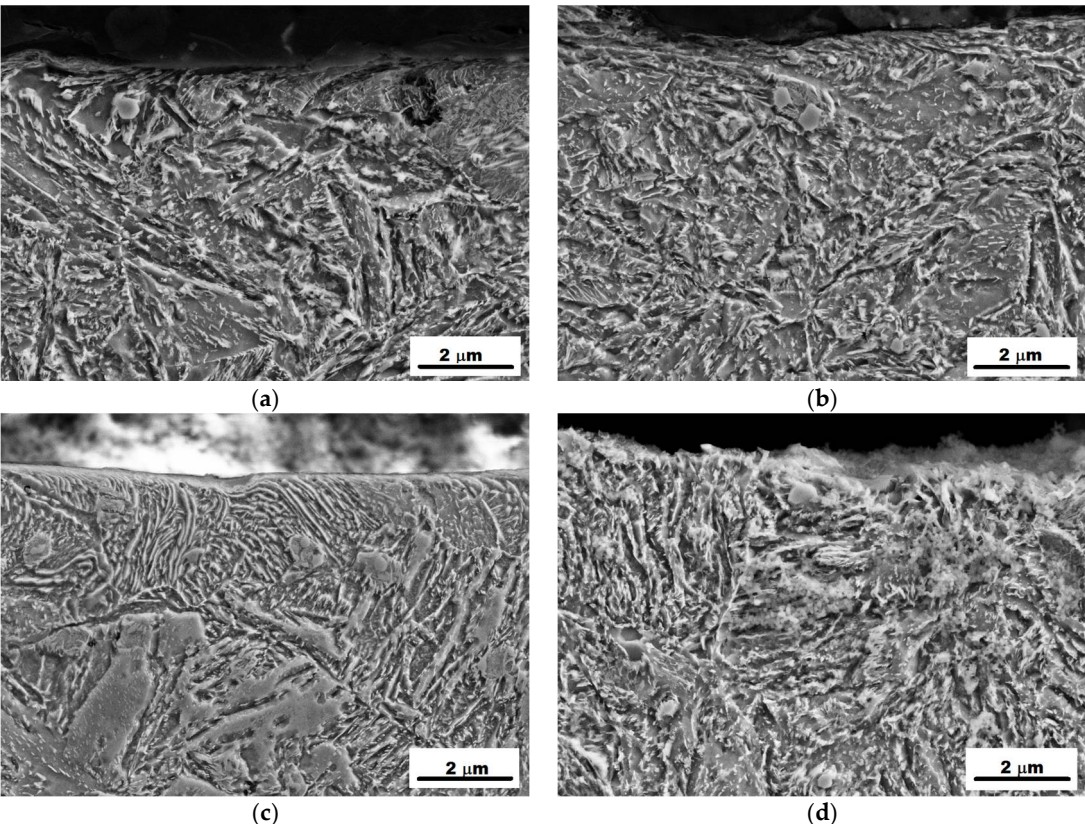

**Figure 3.** Cross sectional images of shot peened surfaces for the different shot peening times. (**a**) 10 s, (**b**) 20 s, (**c**) 40 s, (**d**) 80 s.

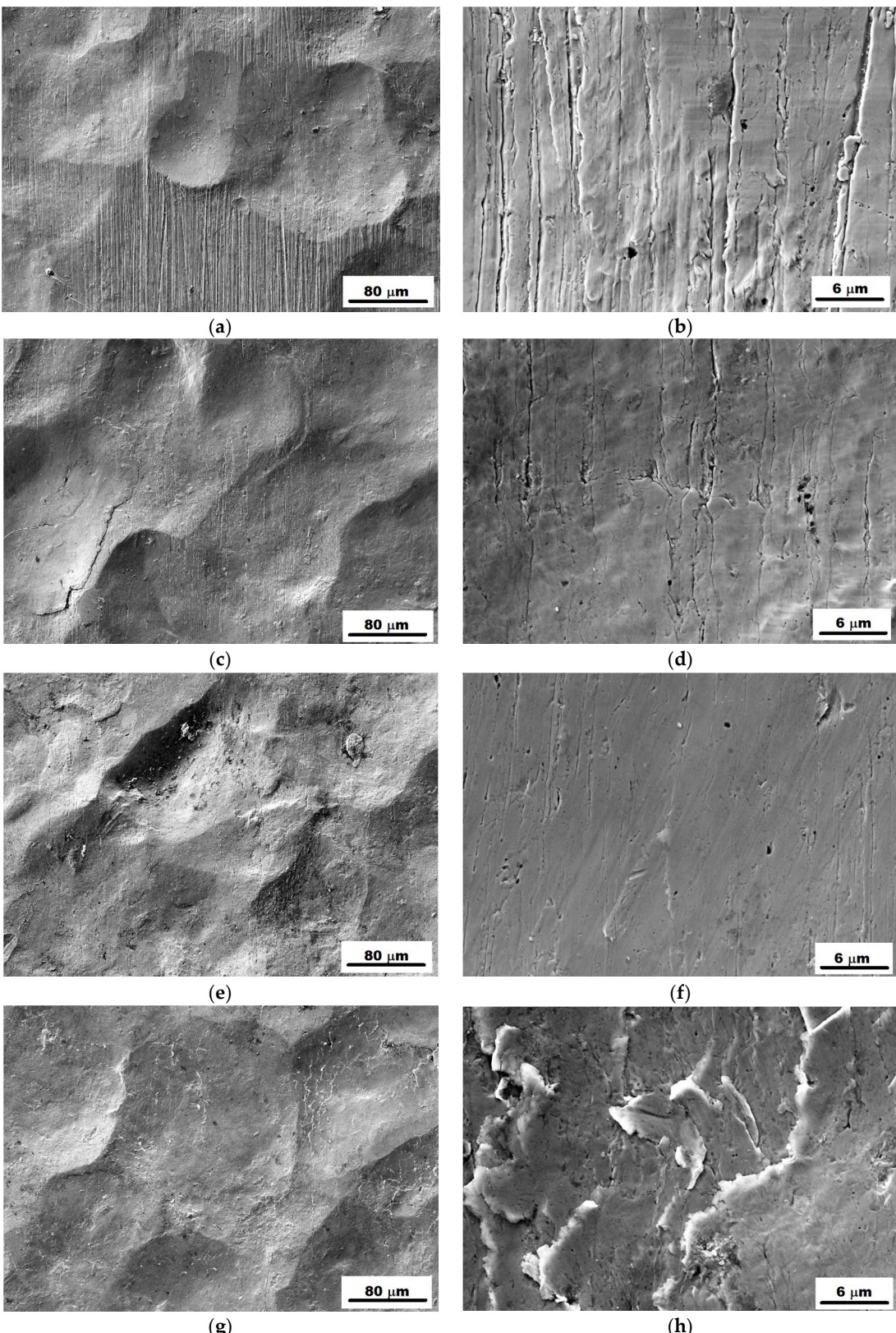

**Figure 4.** Front images of shot peened surfaces for the different shot peening times. (**a**) 10 s, (**b**) 10 s - detail, (**c**) 20 s, (**d**) 20 s - detail, (**e**) 40 s, (**f**) 40 s - detail, (**g**) 80 s, (**h**) 80 s - detail.

Figure 4 also depicts that the produced surface is typical for a shot peening process containing peaks and valleys. The height of those peaks and valleys increases along with the shot peening time

and the corresponding degree of energy accumulated in the shot peened surfaces (see Figure 4, as well as information about surface roughness in Table 2).

### 3.2. XRD Measurements

Compared to CSP [9,10,12], SSP produces stress profiles containing a high magnitude of compressive stresses penetrating quite deep beneath the free surface (see Figure 5). The maximum of these stresses can be found at a depth of 0.1 to 0.2 mm, followed by a steep decrease and change into tensile stresses in the deeper regions. The compressive stresses for the RD and TD directions are quite similar, but TD exhibits about 200 MPa higher magnitudes of compressive stresses. The surface stress for the TD direction is about −600 MPa for all of the SSP regimes. The surface stress for the RD direction is about −390 MPa for all of the shot peening regimes, except for the 40 s regime, which exhibits about 70 MPa higher compressive stresses. The stress profiles for 10 and 20 s (for both directions) are very similar, whereas the 40 s regime produces stress profiles that are gently shifted to the higher stress magnitude. Moreover, the depth in which the maximum of the compressive stresses can be found is higher than those for the 10 and 20 s regimes. The 80 s regime exhibits lower compressive stresses in the descending part of the stress profiles; however, the compressive stresses penetrate deeper than those for the 10 and 20 s regimes (such a statement is also valid for the 40 s regime). The residual stress anisotropy after SP is influenced by the original microstructure of the material after rolling, as well as its significant anisotropy. In addition, the presence of the retained austenite was verified using the XRD technique, however no austenite peak was measured in the irradiated surface, and thus, the presence of retained austenite was excluded.

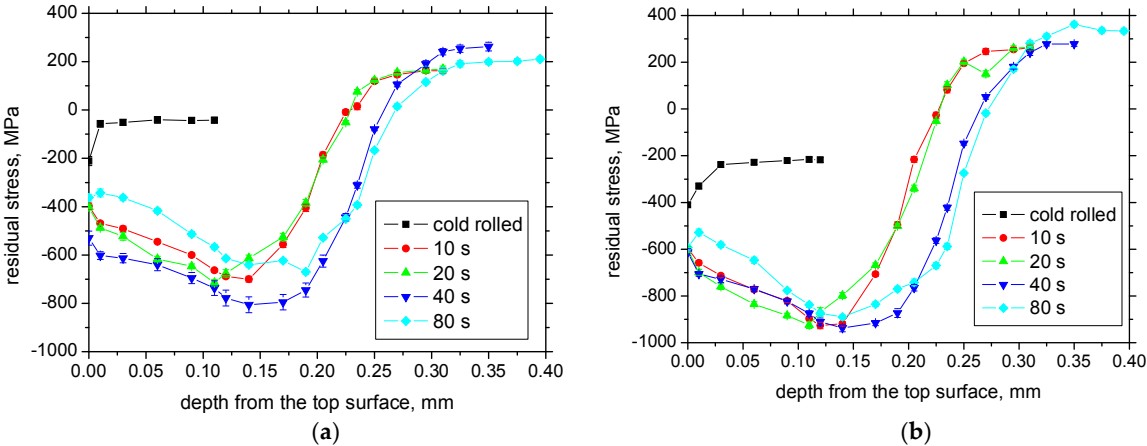

**Figure 5.** Stress profiles for the rolling direction (RD) and transversal direction (TD) directions. (**a**) RD direction, (**b**) TD direction.

The full width at half maximum (FWHM) of the XRD diffraction peaks' profiles are very similar for all of the shot peening regimes, as well as for the RD and TD directions (see Figure 6). The FWHM decreases along with the increasing depth from the free surface. However, the local maximum occurs for all of the FWHM profiles. This maximum can be found at a depth of about 0.20 mm for the 10 and 20 s regimes, whereas it is shifted slightly deeper for the 40 and 80 s regimes. It can be found that this maximum occurs at the depth where the steepest stress gradient can be found (the steepest decrease of compressive stresses).

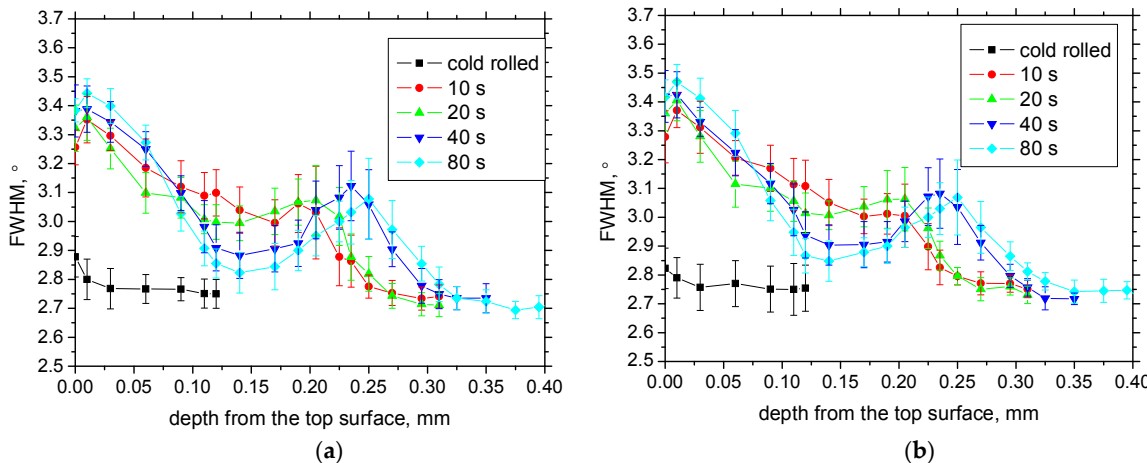

(**a**)                                                          (**b**)

**Figure 6.** The full width at half maximum (FWHM) of XRD for the RD and TD directions. **a**) RD direction, (**b**) TD direction.

### 3.3. MBN Measurements

Figure 7 shows that the MBN increases with the shot peening time. The RD direction exhibits a higher MBN emission than that for the TD direction. As opposed to the evolution of the Almen intensity (exhibits early saturation along with the shot peening time), the MBN increases nearly linearly with the shot-peening time.

The MBN envelopes are shown in Figure 8. An increasing MBN emission is usually associated with an alteration of the stress state and/or the decreasing pinning strength of the magnetizing matrix; therefore, the shift of the MBN envelope to the lower magnetizing field and the corresponding lower magnetic field in which an MBN envelope reaches the maximum (usually referred as PP). However, the MBN envelopes, as well as Figure 9 (PP of the MBN envelopes), clearly demonstrate that the MBN envelopes are shifted to the higher magnetizing fields (only 80 s regime and RD direction exhibits certain decrease).

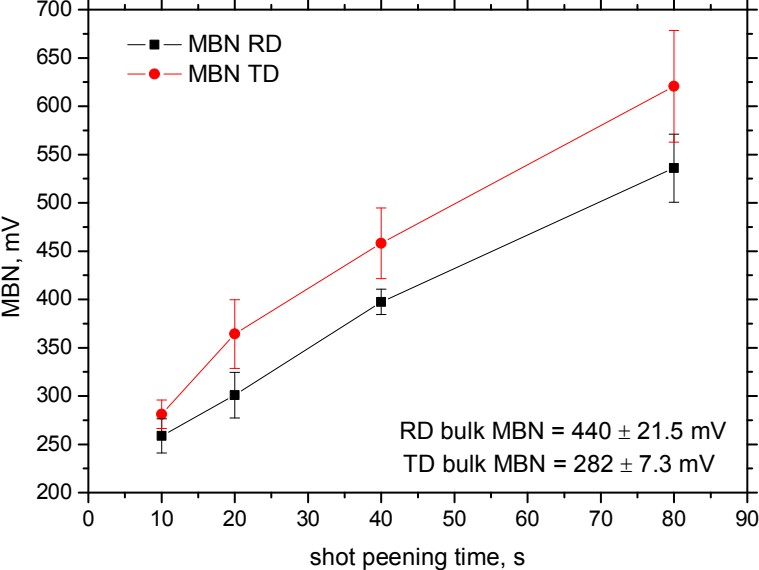

**Figure 7.** MBN (root mean square (RMS) value) after shot peening.

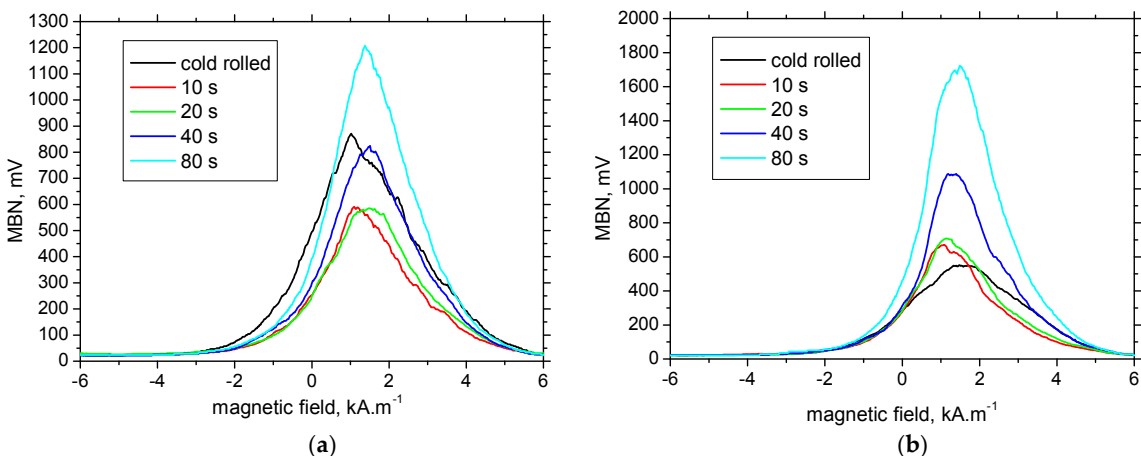

**Figure 8.** MBN envelopes for the RD and TD directions. (**a**) RD direction, (**b**) TD direction.

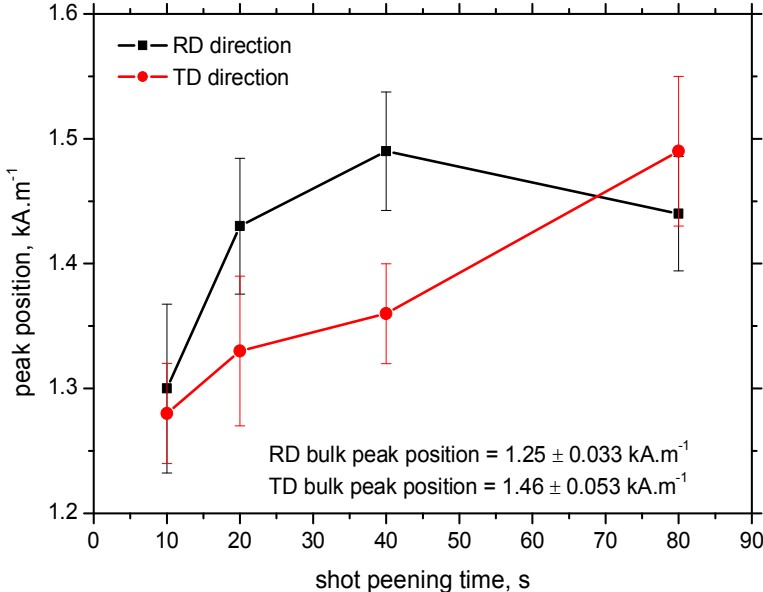

**Figure 9.** The peak position (PP) of the MBN envelopes for the RD and TD directions.

## 4. Discussion of Obtained Results

The evolution of the MBN (as that shown in Figure 7) is controversial with respect to the stress state as well as the accumulation of energy (plastic deformation) with the increasing shot peening time (as shown in Figure 4). Therefore, one might expect that the MBN would decrease with the shot peening time. Taking into consideration the estimated sensing depth of MBN (~70 μm [29]), in this particular case, increasing the magnitude of the compressive stresses with the shot peening time, should contribute to lower MBN emissions [25,26]. The 80 s regime exhibits a certain decrease of residual stresses within the MBN sensing depth (as compared to the other regimes), however such stress differences could not satisfactory and fully explain the much higher MBN for this shot peening time. Furthermore, the TD direction exhibits a higher magnitude of compressive residual stresses, but the MBN for this direction exceeds that for the RD (see Figures 5 and 7). A strong correlation between the residual stresses and the MBN is usually reported for uniaxial elastic stresses. The tensile stresses tend to align the domain walls in the direction of the exerted stress, which, in turn, increases the magnitude of the MBN pulses, whereas decreasing the MBN under compressive stresses is due to the alignment of the domain walls in the direction perpendicular to the direction of the stress [35]. However, under the biaxial or multiaxial stress state, the alignment initiated by the stress in a certain direction has a compensation effect in the other direction [36]. For these reasons, the effect

of the residual stresses on the MBN, in this case, is inferior to the microstructure effects. As Kleber and Vincent [37] reported, the increase of MBN versus plastic strains can only be attributed to the microstructure, especially to the dislocation tangles-domain walls interaction.

The RMS of the MBN signals is defined as follows [38]:

$$\mathrm{RMS} = \sqrt{\frac{1}{n}\sum_{i=1}^{n} X_i^2} \tag{2}$$

where $n$ is the total number of MBN pulses (events) captured at the specific frequency range, and $X_i$ is the amplitude of the individual pulses. The increasing MBN emission, expressed in term of its RMS value, is therefore the result of (i) increasing the number of MBN pulses, (ii) increasing the magnitude of the MBN pulses, and (iii) increasing the number and amplitude of the MBN pulses. Figure 10 depicts that the number of MBN pulses for the RD and TD direction slightly decreases with the shot peening time; thus, the increasing MBN has to be associated with their increasing magnitude. The magnitude of the MBN pulses is mainly driven by the free path of the domain wall motion (domain wall thickness could also contribute) [39]. The accumulation of the energy (deformation) in the shot peened surface with the shot peening time (as Figure 4 clearly illustrates) would increase the dislocation density. Bayramoglu et al. [21] and Ng et al. [40] report that the MBN emission in the case of plastic the deformation depends on two major factors that counteract each other. The first factor is the elongation of the grains, which leads to the reorientation of the domains and the corresponding domain walls, and thus the increasing MBN. The second factor is the presence of the dislocation cells, which hinder the domain wall motion. The higher deformation ratios induce a large number of smaller cell structures, so the boundary area increases, which results in the low MBN emission [32]. The effect of the domains and the corresponding reorientation of the domain walls results in an increase of MBN in the direction of the plastic deformation at the expense of the perpendicular direction. For this reason, this factor in this case does not take place, as the MBN increases in the RD and TD direction.

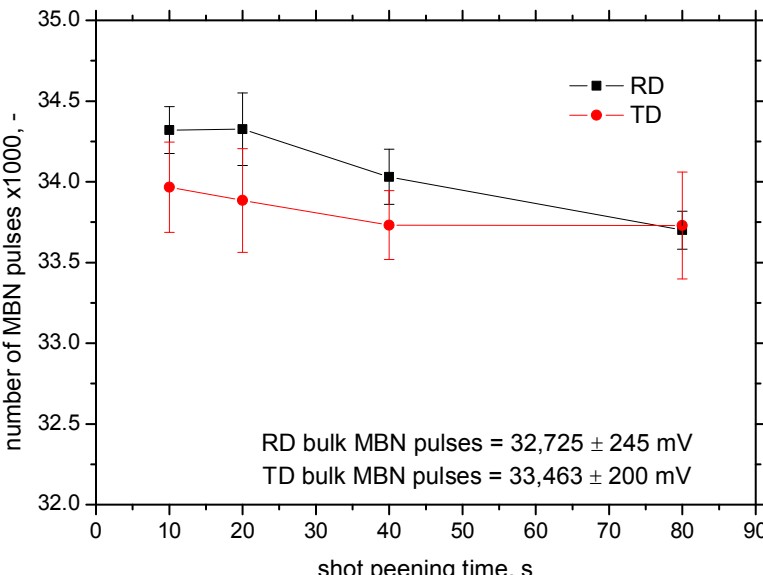

**Figure 10.** Number of detected MBN pulses.

On the other hand, Figure 11 shows the microhardness profiles of the shot peened surfaces. The deeper regions remain untouched by the SSP, whereas the near-the-surface layer exhibits a certain drop in microhardness. It is worth mentioning that the hardness of the strips before (46 ± 2 HRC) and also after shot peening (41 HRC) occurs in the range of the shots hardness (40 ÷ 51 HRC). Such a relationship could explain why the microhardness is not affected more, even though the FWHM of

XRD patterns exhibits quite a high response. In this case, the microhardness strongly correlates with the dislocation density. It is assumed that the shot impacts initiate the dislocation slip in the thin near surface layer.

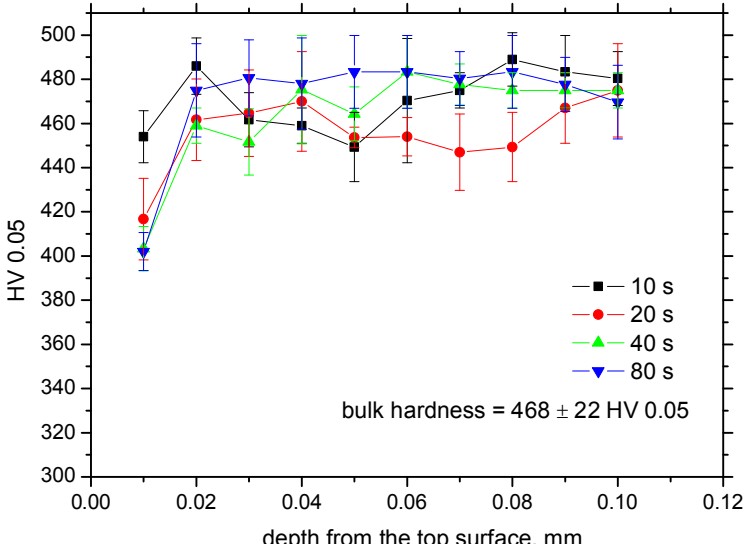

**Figure 11.** Microhardness profiles.

Figure 12 clearly depicts that the average dislocation density, $\rho_D$, gradually increases along with the shot penning time, and exhibits a certain decrease in the shot peening time of 80 s. Such behavior explains the evolution of the PP of the MBN envelope (especially for the RD direction) when the MBN envelopes are shifted to higher magnetic fields along with increasing shot peening time. On the other hand, the increasing MBN is a result of the remarkable redistribution of these dislocations in the near surface region, expressed in terms of the $M_W$ parameter. This parameter presents information about the distribution of dislocations in the matrix, and can be defined as follows [30–34]:

$$M_W = R_c \cdot \sqrt{\rho_D} \tag{3}$$

where $R_c$ is defined as the cut of radius (size of dislocation stress field). Low $M_W$ values indicate a strong non-homogeneity of dislocations density and vice versa.

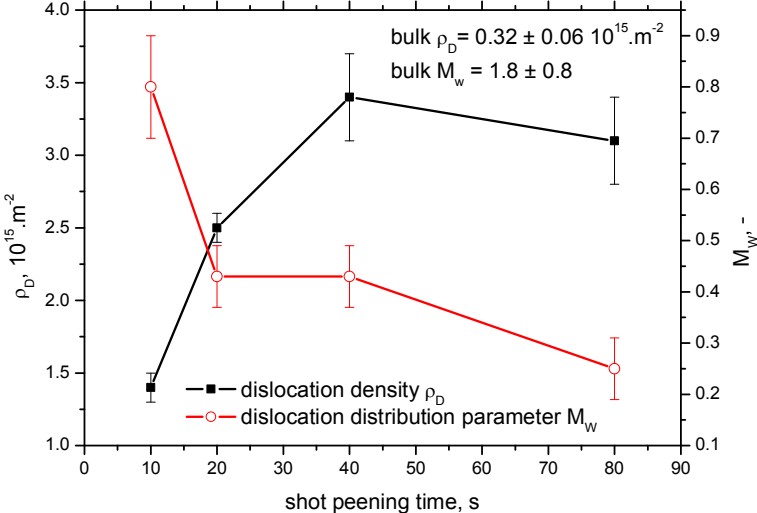

**Figure 12.** Dislocations density and their distribution versus shot peening time.

The size of the coherently diffracting domains (in other words, the crystallite size obtained from the XRD patterns) represents a measure of the average size of the structural units of the material, with the lattice not being distorted by the defects. In the present case, it is estimated that the crystallite size corresponds approximately to the mean size of the dislocation cells, that is, to the regions with a low dislocation density separated by dislocation walls (tangles). In Figure 13, one can see that the size of the coherently scattering domains is higher for the lower microhardness, and vice versa.

The decreasing microhardness (in near-the-surface region), increasing crystallite size, and decreasing $M_W$ with the shot peening time indicate that these aspects play major roles when considering the evolution of MBN. The effective value (RMS) of MBN is mainly driven by the increasing size of the regions containing a low dislocation density, whereas the influence of the dislocation tangles (of high dislocation density) is only minor, see Figure 14. The low $M_W$ indicates that the dislocations are clustered in the dislocation cells and in the matrix containing low dislocation density neighbors with dislocation tangles of very high dislocation density. Figure 12 clearly illustrates that the increasing average dislocation density for the shot peening times of 10, 20 and 40 s is compensated by the increasing non-homogeneity of their distribution (decreasing $M_W$ with the shot peening time). Therefore, the accumulation of the energy of shot impacts (with shot peening time) gradually increases the dislocations density in the dislocation tangles, at the expense of the neighboring cells of a low dislocation density. Figure 13 depicts that the increasing dislocation density in the dislocation tangles is compensated by decreasing the dislocation density of the neighboring cells containing a much lower dislocation density. However, such an evolution tends to be saturated, and the further enlarging of the SP time (80 s regime) results in the gentle decrease of the average dislocation density (see Figure 12). Increasing the dislocation density in the dislocation tangles counteracts against decreasing the dislocation density in the neighboring regions for the 10, 20, and 40 s regimes. MBN values increase with the SP time as a result of the predominating effect of the regions containing the low dislocation density. On the other hand, it is considered that the further increase of MBN for the 80 s regime is due to the synergistic effect of decreasing the dislocation density over the entire grain. Such behavior also confirms the findings of Kleber and Vincent [37], in which a higher MBN is also attributed to the higher mobility of the domain walls in the regions of the low dislocation density, while the high-density regions are much less extended. The crystallite size increases and the surface microhardness decreases versus the shot peening time, as a result of the predominating influence of the cells of a low dislocation density. However, the surface microhardness tends to saturate for longer shot peening times, whereas the crystallite size increases gradually.

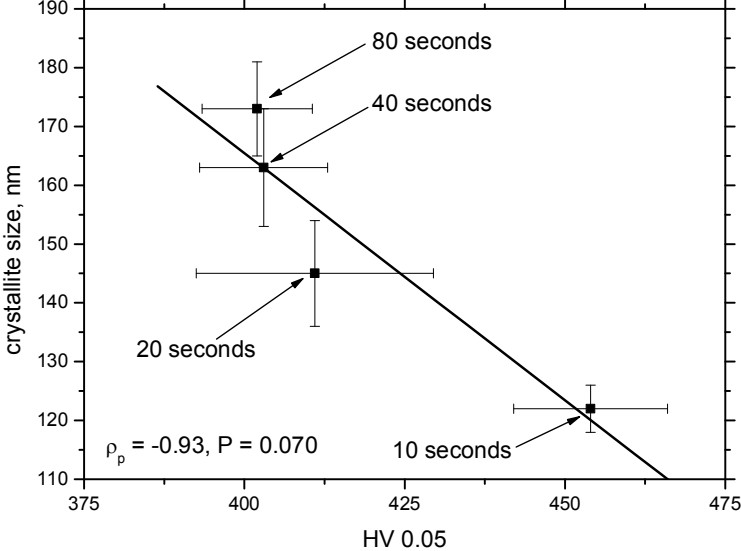

**Figure 13.** Surface microhardness versus crystallite size.

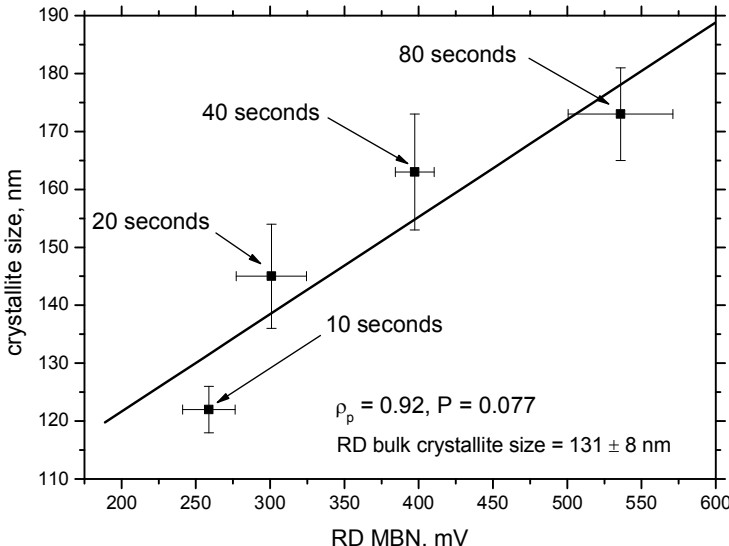

**Figure 14.** MBN (RMS value) versus crystallite size.

## 5. Conclusions

It is worth mentioning that shot peening is very often referred as a process decreasing the MBN, because of the compressive residual stresses and/or surface hardening [25,26]. However, such studies usually report about the CSP or shot peening of samples that behave in a malleable manner. On the other hand, AISI 1070 spring steel is usually employed in the elastic regime of loading, and exhibits poor plastic properties. Thus, the response of the surface to the accumulated shots impacts (in regime of the SSP) and the evolution of MBN against the accumulated energy differ. This study demonstrates a very good correlation between the MBN and the shot peening time, whereas the Almen intensities saturate early. Therefore, the MBN technique could be employed for the non-destructive evaluation of the surface after the SSP.

**Author Contributions:** Conceptualization, M.N., L.T., and J.B.; methodology, M.N. and L.T.; software, M.N., J.Č. (Jakub Čížek), J.Č. (Jiří Čapek), and J.M.; validation, L.T., F.P., and J.M.; formal analysis, J.B. and J.M.; investigation, J.Č. (Jiří Čapek), P.M., M.N., and F.P.; resources, J.B. and J.M.; data curation, M.N. and L.T; writing (original draft preparation), M.N. and L.T.; writing (review and editing), visualization, J.B. and J.M.; supervision, M.N. and L.T.; project administration, M.N.; funding acquisition, M.N., L.T., and P.M.

**Funding:** This study was supported by APVV project no. 16-0276 and 14-0284, VEGA project no. 1/0121/17, and KEGA project no. 008ŽU-4/2018. P.M. acknowledges the financial support of the Czech Science Foundation under project 14-36566G, and J.Čí acknowledges the financial support of the Czech Science Agency project P108/12/G043.

**Conflicts of Interest:** The authors declare no conflict of interest.

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
