# Peer review of "Non-Destructive Evaluation of Steel Surfaces after Severe Plastic Deformation via the Barkhausen Noise Technique"

_metals, doi:10.3390/met8121029_

Round 1

Reviewer 1 Report

REFEREE COMMENTS:

Non - destructive evaluation of steel surfaces after severe plastic deformation via Barkhausen noise  technique

In this paper severe shot peening (SSP) was studied. The studies focused on the XRD and Barkhausen noise studies. The paper states the need to evaluate the SP outcome in nondestructive way and points out the complex case of deformed surface structure to the BN features.

Introduction

You point out the difference of the CSP and SSP but in the introduction section it is still a bit unclear where the SSP treatment is utilized and what is the purpose of the study therefore? Just to differentiate the SSP from CSP. Should SSP changes be seen in the cross sectional microstructure figures (what have other studies showed)?

Materials and methods

-Figure 1: Almen intensity lacks the unit, check also Table 2. In my opinion in the standard regarding the Almen intensity SAE J443 the unit was mm A.

-You could add the shot hardness in the details given prior Figure 1. And also state was the sample surface harder or softer than the shots prior the shot peening?

-Could you add the yield stress for AISI1070?

Results

-In Fig 3 a), b) the scale bars are cut so that they are not visible

-Could figures 4 and 5 be joined next to each other as 4a+b to illustrate better the difference in RD and TD? Also figures 6 and 7?

-p.7 line 166-167: You state that the SSP produces stress profiles penetrating quite deep beneath. In here you could refer to other studies to indicate that the CSP really produces half or less the deformed layer?

-p.7-8, lines: -> also p.13, line 263-4: The change in shot peening time 80s behavior is interesting and you explain it with the decrease of dislocation density. Do you have any suggestion why the behavior changes from 40s to 80s? There is also a change in Fig 6&7 in position 0.25 mm for the 40s and 80s samples (and as you explained in the region with the steepest decrease of compressive RS).

-p.9: What added value does the figures 8 and 9 bring to the article? If they are left then they should be joined also next to each other more clearly to point out the differences in BN burst with 10s and 80s. However, the values of MBN are already given in Fig 10 so I would suggest to left figures 8 and 9 out.

-p.10-11, Add figures 11 and 12 next to each other, then it is more easy to compare the RD and TD directions.

Discussion

-p.11, line 221-228: Have you checked this paper by Kleber and Vincent (X. Kleber, A. Vincent, On the role of residual internal stresses and dislocations on Barkhausen noise in plastically deformed steel, NDT&E International, 37 (2004) 439–445.)? In their studies plastic deformation with plastic strain more than 1% affected that the BN activity was increased. (Also issues of critical stress and induced stress anisotropy..)

And also you probably need to point out the BN measurement depth issue in the discussion. Because due to the Rs gradient, the BN measurement depth is on certain depth and what is the order of the samples with different peening times on that point? May the high compression RS affect to the obtained measurement depth (increasing it) that we are measuring on the region where 80s peening time has more ?

-Fig.15 is the depth scale (µm) correct in the Figure 15? It should be pointed out the hardness relation of the shots and the AISI1070 material. It is surprising that the microhardness is not affected more even though the FWHM XRD shows higher response.

-What happens in the region from 40s to 80s (dislocations/other issues)? I think you did not conclude the changing effect much in the discussion part.

-Fig 17. Could the shot peening times be linked to the measurement points? Does the 80 s peening time show different behavior?

Also the different behavior in TD/RD direction for RS? The XRD FWHM shows that the peak broadening is similar to both directions. Can the direction of shot peening affect the directionality of the dislocation cell structure?

Did you verify that there is no de-carburized layer after the heat treatment in the surface layer? The strips were really thin, 1.2mm, could there be also similar responses to same material but thicker samples?

How would you verify your assumptions from the dislocation structure changes? Could TEM or other means be used?

Author Response

All changes made in the manuscript (additional texts and corrections) are highlighted yellow color (valid for the manuscript as well as this document).

Reviewer n. 1:

In this paper severe shot peening (SSP) was studied. The studies focused on the XRD and Barkhausen noise studies. The paper states the need to evaluate the SP outcome in nondestructive way and points out the complex case of deformed surface structure to the BN features.

Reviewer: You point out the difference of the CSP and SSP but in the introduction section it is still a bit unclear where the SSP treatment is utilized and what is the purpose of the study therefore? Just to differentiate the SSP from CSP.

Response: We agree and therefore we added explanation – can be found at the end of the chapter “Introduction”.

Manuscript: There is no exact boundary between the CSP and SSP treatments, because it significantly depends on the type of the treated material. For example, SSP parameters for steels are much more severe than for alumínium alloys. However, the CSP process is mainly connected with accumulation of residual stresses in the subsurface volume, while the SSP treatment is considered when besides the residual stresses significant changes in the microstructure (mainly grain refinement) occurs. For these reasons the magnetic properties and the corresponding MBN emission (due to superimposing effects of the microstructure and residual stresses alterations) after SSP should be affected, as well. This is a case study focused on SSP surfaces obtained via the different regimes (the corresponding Almen intensities) in which the residual stresses and microstructure influence on MBN emission is investigated.

Reviewer: Should SSP changes be seen in the cross sectional microstructure figures (what have other studies showed)?

Response: We carried out the metallographic as well as SEM observations of shot peened surfaces. SEM scans (illustrated in Fig. 2) as well as metallographic images exhibited no remarkable changes (we applied 1% concentration of Nital for 10 seconds - application of higher concentrations or/and longer etching time resulted into over etching of the shot peened surfaces). Please see metallographic images. The main reason is probably associated with the matrix before shot peening process. AISI steel 1070 is quenches and subsequently high tempered. Therefore this matrix behaves more in elastic regime and exhibits poor plastic properties and higher resistance against remarkable structure transformations. For these reasons, it was really difficult for us for quite long time to explain evolution of MBN versus shot peening time and we could not understand and explain MBN increase with peening time. 

Shot   peening time 40 seconds

Shot peening   time 80 seconds

Metallographic images of shot peened surfaces, Nital 1% for 10 seconds

Manuscript: we prefer no changes

Reviewer: Figure 1: Almen intensity lacks the unit, check also Table 2. In my opinion in the standard regarding the Almen intensity SAE J443 the unit was mm A.

Response: we added units - mm A

Manuscript: added A and mm A - please check Table 2 ad Fig. 1.

Reviewer: You could add the shot hardness in the details given prior Figure 1. And also state was the sample surface harder or softer than the shots prior the shot peening?

Response: shot hardness in the range from 40 to 51 HRC. Hardness of the samples before shot peening is 468±22 HV which corresponds 46±2 HRC. Hardness due to shot peening in the near surface layer decrease and softer surface exhibits approx. 400 HV (approx. 41 HRC).

Manuscript: The shots (S170) hardness occurs in the range from 40 to 51 HRC whereas the strips hardness is 468±22 HV which corresponds 46±2 HRC.

Reviewer: Could you add the yield stress for AISI1070?

Response: added the required information

Manuscript: check it in the chapter 2 - tensile strength 640 MPa, yield strength 495 MPa,

Reviewer: In Fig 3 a), b) the scale bars are cut so that they are not visible

Response: added new scales

Manuscript: please check appearance of Fig. 2 and Fig. 3

Reviewer: Could figures 4 and 5 be joined next to each other as 4a+b to illustrate better the difference in RD and TD? Also figures 6 and 7?

Response: corrected as required

Manuscript: please check it in the manuscript – new order of Figures and changes in the corresponding text

Reviewer: p.7 line 166-167: You state that the SSP produces stress profiles penetrating quite deep beneath. In here you could refer to other studies to indicate that the CSP really produces half or less the deformed layer?

Response: we added references to the other studies as indicated below.

Manuscript: As compared to the CSP [9, 10, 12], the SSP produces stress profiles containing……

Reviewer: p.9: What added value does the figures 8 and 9 bring to the article? If they are left then they should be joined also next to each other more clearly to point out the differences in BN burst with 10s and 80s. However, the values of MBN are already given in Fig 10 so I would suggest to left figures 8 and 9 out.

Response: we agree and these figures (as well as the corresponding text) were removed from the manuscript

Manuscript: figures 8 and 9 removed from the manuscript and new order of figures edited.

Reviewer: p.10-11, Add figures 11 and 12 next to each other, then it is more easy to compare the RD and TD directions.

Response: corrected as required

Manuscript: please check it in the manuscript – new order of Figures and changes in the corresponding text

Reviewer: p.11, line 221-228: Have you checked this paper by Kleber and Vincent (X. Kleber, A. Vincent, On the role of residual internal stresses and dislocations on Barkhausen noise in plastically deformed steel, NDT&E International, 37 (2004) 439–445.)? In their studies plastic deformation with plastic strain more than 1% affected that the BN activity was increased. (Also issues of critical stress and induced stress anisotropy..)

Response: Thank you for your link. We knew about this study and we also think that reviewer link is helpful for us since this study really explains increasing MBN along with increasing plastic strains. We added explaining text in chapter “Discussion of obtained results”.

Manuscript: As Kleber and Vincent [37] reported, increase of MBN versus plastic strains can only be attributed to the microstructure, especially dislocation tangles-domain walls interaction.

Such behaviour confirms also findings of Kleber and Vincent [37] in which the higher MBN are also attributed to the higher mobility of domain walls in the regions of the low dislocation density, while the high density regions are much less extended. 

Reviewer: And also you probably need to point out the BN measurement depth issue in the discussion. Because due to the RS gradient, the BN measurement depth is on certain depth and what is the order of the samples with different peening times on that point? May the high compression RS affect to the obtained measurement depth (increasing it) that we are measuring on the region where 80s peening time has more?

Response: Of course, analyzing the influence of stresses on MBN we took into the consideration the estimated sensing depth of MBN but it was not emphasized in the manuscript. Therefore we made minor correction in the associated text as it is indicated below (see chapter “Discussion of obtained results”). On the other hand, we think that influence of residual stresses on MBN (taking MBN sensing depth into consideration) in this case is only minor and microstructure dominates. The 80 seconds regime exhibits certain decrease of RS within MBN sensing depth (as compared to the other regimes), however such stress differences could not satisfactory and fully explain much higher MBN for the 80 seconds regime. 

Manuscript: Taking into consideration the estimated sensing depth of MBN (~70 mm [29]) in this particular case, increasing magnitude of compressive stresses with the shot peening time should contribute to the lower MBN emission [25, 26]. The 80 seconds regime exhibits certain decrease of residual stresses within the MBN sensing depth (as compared to the other regimes), however such stress differences could not satisfactory and fully explain much higher MBN for this shot peening time.

Reviewer: Fig.15 is the depth scale (µm) correct in the Figure 15? It should be pointed out the hardness relation of the shots and the AISI1070 material. It is surprising that the microhardness is not affected more even though the FWHM XRD shows higher response.

Response: reviewer is right – the depth scale was corrected. Many thanks for your idea about relation between shots hardness and measured microhardness. We found it important and reasonable considering data interpretation. Moreover, it would be interesting topic for the next study.

Manuscript: we added hardness relations as well as the associated explanation – see chapter “Discussion of obtained results”.

It is worth to mention that the strips hardness before (46±2 HRC) and also after shot peening (41 HRC) occurs in the range of shots hardness (40÷51 HRC). Such relation could explain why microhardness is not affected more even though the FWHM of XRD patterns exhibits quite high response.

Reviewer: p.7-8, lines: -> also p.13, line 263-4: The change in shot peening time 80s behavior is interesting and you explain it with the decrease of dislocation density. Do you have any suggestion why the behavior changes from 40s to 80s? There is also a change in Fig 6&7 in position 0.25 mm for the 40s and 80s samples (and as you explained in the region with the steepest decrease of compressive RS).

Reviewer: What happens in the region from 40s to 80s (dislocations/other issues)? I think you did not conclude the changing effect much in the discussion part.

Response: Your remark is very beneficial for us and we had to think about it. Many thanks for indication of this matter. We added more detail explanation are indicated below.

Manuscript: Figure 12 clearly illustrates that the increasing average dislocation density for shot peening times 10, 20 and 40 seconds is compensated by the increasing non-homogeneity of their distribution (decreasing MW with the shot peening time). Therefore, accumulation of energy of shot impacts (with shot peening time) gradually increase the dislocations density in the dislocation tangles at the expense of neighbouring cells of low dislocation density. Figure 13 depicts that increasing dislocation density in the dislocation tangles is compensated by decreasing the dislocation density of neighbouring cells containing much lower dislocation density. However, such evolution tends to saturate and the further enlarging of SP time (80 seconds regime) results into the gentle decrease of the average dislocation density (see Figure 12). Increasing dislocation density in the dislocation tangles is counteracting against decreasing dislocation density in the neighbouring regions for 10, 20 and 40 seconds regimes. MBN values increase with the SP time as a result of the predominating effect of the regions containing the low dislocation density. On the other hand, it is considered that the further increase of MBN for 80 seconds regime is due to the synergistic effect of decreasing dislocation density over the entire grain.  

Reviewer: Fig 17. Could the shot peening times be linked to the measurement points? Does the 80 s peening time show different behavior?

Response: we tried to add the additional x –axe (with shot peening time) to this figure but we failed applying linear as well as logarithmic scales. For this reason we only added indications for all measured points as it is illustrated in the Figure – please check the appearance of the figure.

Manuscript: (certain explanation is reported in the previous item) but we also added the following text - The crystallite size increases and surface microhardness decreases versus shot peening time as a results of predominating influence of the cells of low dislocation density. However, the surface microhardness tends to saturates for longer shot peening times whereas the crystallite size increases gradually.

Reviewer: Also the different behavior in TD/RD direction for RS? The XRD FWHM shows that the peak broadening is similar to both directions. Can the direction of shot peening affect the directionality of the dislocation cell structure?

Response: The direction of shot peening does not affect the directionality of dislocation cell structure because the character of the technology performed at angle perpendicular to the surface created the isotropic surface from this point of view. However, the residual stress values are influenced by the original microstructure of the material after rolling and its significant anisotropy.

Manuscript: The residual stress anisotropy after SP is influenced by the original microstructure of the material after rolling and its significant anisotropy.

Reviewer: Did you verify that there is no de-carburized layer after the heat treatment in the surface layer? The strips were really thin, 1.2mm, could there be also similar responses to same material but thicker samples?

Response: As it is indicated before, we carried out investigation of microstructure via metallographic as well as SEM observations and we did not observe de-carburized layers – please see microstructure illustrated in the aforementioned list. We are not able to provide information about response considering thicker samples since we do not have such samples. However, it is considered that changes in microstructure would be similar only certain differences could be expected in residual stresses profiles.

Manuscript: we prefer no changes

Reviewer: How would you verify your assumptions from the dislocation structure changes? Could TEM or other means be used?

Response: Yes probably TEM would be possible to identify the dislocation structure changes, however it is extremely difficult to prepare TEM specimens from the SSP layers on steel (magnetic) material and it is beyond the purpose of this study.

Manuscript: we prefer no change

Reviewer 2 Report

The article describes the characterisation of the shot peening process of Almen strips by means of Almen intensity measurements, XRD, hardness and Barkhausen noise measurements. The motivation of the study is clear, the shot peening process and the material parameter relating thereto should be characterised non-destructively by the means of Barkhausen noise measurements.

The study has a very broad approach, with different characterisation methods. The manuscript shows some structural flaws. The author should differentiate between results and discussion. The results for MBN impulses, micro-hardness measurements were introduced in the discussion chapter.

Line 114: common abbreviation for root mean square is RMS (abbreviations should be introduced by the first appearance)

Figure 1: Unit for Almen intensity should be added

Tabel 2: Unit for Almen intensity should be added

                More information about the peening parameter should be added (flowrate, pressure …)

                Were the MBN and XRD measurements carried out on flat or on deformed Almen strips?

                A picture of the experimental setup (Shot peening/MBN/XRD) would increase the comprehensibility significantly

Figure 2/3: scale bars should be significantly larger

                The captions should be more precise

Figure 6/7: unit for FWHM should be added

Line 187: FWHM is not introduced

Figure 8/9: caption should be more precise

Figure 10: MBN should be named max. of MBN envelope (difference to MBN in Fig 8/9 is not clear)

Line 228: BW is not introduced

Line 263: It is not stated how the dislocation density and distribution was determined

Recommend further publications for integration in “References”:

Santa-aho, S.; Vippola, M.; Sorsa, A.; Leiviskä, K.; Lindgren, M.; Lepistö, T.:
Utilization of Barkhausen noise magnetizing sweeps for case-depth detection from hardened steel. NDT&E Int. 2012, 52, 95–102.

Baak, N.; Garlich, M.; Schmiedt, A.; Bambach, M.; Walther, F.:
Characterization of residual stresses in austenitic disc springs induced by martensite formation during incremental forming using micromagnetic methods.
Mater. Test. 2017, 59, 309–314.

Author Response

The article describes the characterization of the shot peening process of Almen strips by means of Almen intensity measurements, XRD, hardness and Barkhausen noise measurements. The motivation of the study is clear, the shot peening process and the material parameter relating thereto should be characterized non-destructively by the means of Barkhausen noise measurements.

Reviewer: The study has a very broad approach, with different characterization methods. The manuscript shows some structural flaws. The author should differentiate between results and discussion. The results for MBN impulses, micro-hardness measurements were introduced in the discussion chapter.

Response: reviewer is right. We spend a lot of time discussing about appearance of chapter “Discussion of obtained results”. This chapter critically discusses evolution of MBN (especially unexpected increase of MBN versus shot peening time) as a function of two main aspects associated with SSP such as stress state and microstructure. Therefore we preferred to present information about MBN pulses as well as microstructure in this chapter instead of the previous chapter (since for reasonable explanation these figures and the corresponding text are important). We understand your point of view, but we would prefer to remain the structure as it is. Please reconsider this aspect. 

Manuscript: we prefer no change.

Reviewer: Line 114: common abbreviation for root mean square is RMS (abbreviations should be introduced by the first appearance)

Response: rms abbreviation explained in chapter 2

Manuscript: The MBN refers to the root mean square (rms) value of the signal.

Reviewer: Figure 1: Unit for Almen intensity should be added, Table 2: Unit for Almen intensity should be added

Response: we added units - mm A

Manuscript: added mm A - please check Table 2 ad Fig. 1.

Reviewer: More information about the peening parameter should be added (flowrate, pressure …)

Response: The only two parameters which are correctly provided for shot peening process is Almen intensity and Coverage. The reason is, that nozzle pressure, flowrate depends on the shot peening machine, however identical peening results can be obtained. The Almen intensity integrates the kinetic energy of the shot stream and by performing treatments with identical Almen intensity, you receive identical surface layers. E. g. high pressure and long distance of the nozzle from the surface can provide same layer as lower pressure from smaller distance…and so on.

Manuscript: we prefer no change

Reviewer: Were the MBN and XRD measurements carried out on flat or on deformed Almen strips?

Response: measurements were carried out on the deformed Almen strips

Manuscript:  we added the sentence (chapter 2. Materials and methods) - The measurements were carried out on the deformed Almen strips.

Reviewer: A picture of the experimental setup (Shot peening/MBN/XRD) would increase the comprehensibility significantly

Response: we added the brief sketch of experiments as they were consequently carried out. 

Manuscript: The brief sketch of the analyses as they were consequently carried out is illustrated in Figure 2. First phase represents non-destructive measurements followed by the XRD stress profile investigations. Consequently, the shot peened strips were cut and hot moulded for SEM observations and microhardness measurements.

Figure 2. Brief sketch of experimental setup.

Reviewer: Figure 2/3: scale bars should be significantly larger. The captions should be more precise

Response: added new scales + captions

Manuscript: please check appearance of Fig. 2 and Fig. 3 and their captions, added - for the different shot peening times.

Reviewer: Figure 6/7: unit for FWHM should be added

Response: corrected

Manuscript: units in figures 6 and 7 added – please check it in the manuscript (but take into consideration the new figures numbering order due to the requirements of reviewer 2)

Reviewer: Line 187: FWHM is not introduced

Response: FWHM abbreviation explained

Manuscript: check it in chapter 3.2. - The FWHM (Full Width at Half Maximum) of XRD diffraction peaks profiles ……

Reviewer: Figure 8/9: caption should be more precise

Response: on the base of reviewer 1 recommendation these figures were removed from the manuscript

Manuscript: figures 8 and 9 (as well as the corresponding text) removed from the manuscript and new order of figures edited.

Reviewer: Figure 10: MBN should be named max. of MBN envelope (difference to MBN in Fig 8/9 is not clear)

Response: corrected

Manuscript: Figure 6. MBN (rms value) after shot peening.

Reviewer: Line 228: BW is not introduced

Response: we replaced abbreviation BW with full text “domain walls”

Manuscript: check it in the chapter 4 domain walls – domain walls

Reviewer: Line 263: It is not stated how the dislocation density and distribution was determined

Response: this information is introduced in chapter 2. Materials and Methods - “dislocation density D were determined from the XRD patterns obtained on an X’Pert PRO diffractometer using the CoKα radiation and subsequent Rietveld refinement performed in Mstruct software [28 – 32] (used for the fitting of the measured XRD patterns).“

Manuscript: no change

Reviewer: Recommend further publications for integration in “References”:

Santa-aho, S.; Vippola, M.; Sorsa, A.; Leiviskä, K.; Lindgren, M.; Lepistö, T.:
Utilization of Barkhausen noise magnetizing sweeps for case-depth detection from hardened steel. NDT&E Int. 2012, 52, 95–102.

Baak, N.; Garlich, M.; Schmiedt, A.; Bambach, M.; Walther, F.: 
Characterization of residual stresses in austenitic disc springs induced by martensite formation during incremental forming using micromagnetic methods. Mater. Test. 2017, 59, 309–314.

Response: we integrated the aforementioned publication in the manuscript main body as well as “references”

Manuscript: please check references 18 and 19 in the list of References

It is well known that the higher MBN can be obtained under tensile stresses than under the compressive ones [14 ‒ 19].

Round 2

Reviewer 1 Report

The manuscript is now accepted after the corrections.

Author Response

Thanks for your appreciation and kind help.

Reviewer 2 Report

The authors corrected the description of the tables and figures and introduced all abbreviations. Moreover, the presentation of the experimental setup is now clearer. Nevertheless, the structure is still very unconventional, even when the authors purposely showed some results in the discussion chapter.

Table 2: "A" should not be written after every number, unit was correctly added in the first row; the numbers seem to be very high

Figure 2: An illustration of the test setup was requested not of the chronology. E.g. application of sensor on the strips,…

Figure 3/4: peened not penned

Figure 14: MBN is the RMS of MBN?

Regarding aforementioned comments:

Reviewer: Were the MBN and XRD measurements carried out on flat or on deformed Almen strips?

Response: measurements were carried out on the deformed Almen strips

Manuscript:  we added the sentence (chapter 2. Materials and methods) - The measurements were carried out on the deformed Almen strips.

Would MBN and XRD measurements on undeformed Almen strips (shot peened, but still in clamping aperture) lead to more significant measurements?

Author Response

Reviewer: Table 2: "A" should not be written after every number, unit was correctly added in the first row; the numbers seem to be very high

Response: corrected

Manuscript: please check Table 2

Reviewer: Figure 2: An illustration of the test setup was requested not of the chronology. E.g. application of sensor on the strips,…

Response: new illustration edited

Manuscript: please check appearance of Fig. 2.

Reviewer: Figure 3/4: peened not penned

Response: corrected

Manuscript: please check it in the manuscript

Reviewer: Figure 14: MBN is the RMS of MBN?

Response: yes – added information

Manuscript: Figure 14. MBN (rms value) versus crystallite size.

Reviewer: Would MBN and XRD measurements on undeformed Almen strips (shot peened, but still in clamping aperture) lead to more significant measurements?

Response: Such measurement would affect only stress state. However, we are not able to evaluate the difference between clamped and free strips now.

Manuscript: no change
